

**ASSESSING THE PEATLAND HUMMOCK-HOLLOW CLASSIFICATION**
**FRAMEWORK USING HIGH-RESOLUTION ELEVATION MODELS: IMPLICATIONS**
**FOR APPROPRIATE COMPLEXITY ECOSYSTEM MODELLING**
Paul A. Moore[1*], Maxwell C. Lukenbach[1], Dan K. Thompson[2], Nick Kettridge[3], Gustaf
Granath[4], and James M. Waddington[1]
[1] School of Geography and Earth Sciences, McMaster University,1280 Main Street West,
Hamilton, ON, L8S 4K1, Canada
[2] Northern Forestry Centre, Canadian Forest Service, Natural Resources Canada,
Edmonton, Alberta, AB, T6H 3S5, Canada
[3] School of Geography, Earth and Environmental Sciences, University of Birmingham,
Edgbaston, Birmingham, B15 2TT, United Kingdom.
[4] Department of Ecology and Genetics, Uppsala University, Norbyvägen 18D, 736 52
Uppsala, Sweden
* Corresponding author: Paul Moore (paul.moore82@gmail.com)
Manuscript for submission to Biogeosciences
KEY WORDS: peatland, microtopography, morphometry, hummock, hollow, sampling
design, ecosystem modelling, digital elevation model



## ABSTRACT

The hummock-hollow classification framework used to categorize peatland ecosystem microtopography is pervasive throughout peatland experimental designs and current peatland ecosystem modelling approaches. However, identifying what constitutes a representative hummock-hollow pair within a site and characterizing hummock-hollow variability within or between peatlands remains largely unassessed. Using structure-from-motion (SfM), high resolution digital elevation models (DEM) of hummock-hollow microtopography were used to: 1) examined how much area needs to be sampled to characterize site-level microtopographic variation; and 2) examine the potential role of microtopographic shape/structure on biogeochemical fluxes using data from 9 norther peatlands. To capture 95% of site-level microtopographic variability, on average an aggregate sampling area of 32 m$^2$ composed of ten randomly located plots with vegetation removed was required. We further present non-destructive transect-based results as an alternative to the SfM approach. Microtopography at the plot-level was often found to be non-bimodal, as assessed using a Gaussian mixture model (GMM). Our findings suggest that the non-bimodal distribution of microtopography at the plot-level may result in an under-sampling of intermediate topographic position. Extended to the modelling domain, an under-representation of intermediate microtopographic positions is shown to lead to large flux biases over a wide range of water table positions for ecosystem processes which are non-linearly related to water and energy availability at the moss surface. A range of tools examined herein can be used to easily parameterize peatland models, from GMMs used as simple transfer functions, to spatially explicit fractal landscapes based on simple power law relations between





microtopographic variability and scale.
**INTRODUCTION**
Northern peatlands in the maritime-temperate, boreal, and subarctic have been
persistent terrestrial sinks for carbon throughout the Holocene, storing approximately
one-third of all global soil carbon (Yu, 2012). However, these peatland carbon stores
are now considered to be at risk from the effects of climate change due to warmer
temperatures and prolonged periods of drought which would increase carbon loss
through decomposition and increased wildfire consumption (Moore et al., 1998; Yu et al.,
2009; Turetsky et al., 2002; Kettridge et al., 2015). While these positive feedbacks
cause carbon loss (*e.g.* Ise et al., 2008; Blodau et al., 2004), the long-term stability of
peatland carbon may be maintained by negative ecohydrological feedbacks that
promote resilience to environmental change (Belyea and Clymo, 2001; Waddington et
al., 2015; Hodgkins et al., 2018). These negative feedbacks depend, in part, on the
presence of microtopography (microforms) that provides spatial diversity in
ecohydrological structure and biogeochemical function across a peatland (Belyea and
Clymo, 2001; Belyea and Malmer, 2004; Eppinga et al., 2008; Pedrotti et al., 2014;
Malhotra et al., 2016).

Peatland microform classification is typically defined by their proximity to the water table
and characteristic vegetation assemblages, such as different species of *Sphagnum*
moss and cover of woody shrubs (Andrus et al., 1983; Rydin and McDonald, 1985;
Belyea and Clymo, 1998). Hummocks and hollows occur at a spatial scale of 1 to 10 m



(S2 – Belyea and Baird, 2006), with the surface of hummocks usually covering an area
on the order of 1 m². The hummock surface is typically located ~0.20 m or higher above
the water table (Belyea and Clymo 1998; Malhotra et al., 2016). Hollows are closer to
the water table and may occasionally be inundated, and 'lawns' are intermediate to
hummocks and hollows (Belyea and Clymo, 1998).

Conceptualizing and qualitatively classifying complex peatland microtopography as
hummocks and hollows is common in peatland research (*e.g.* Waddington and Roulet
1996; Belyea and Clymo 2001; Nungesser 2003; Benscoter et al., 2005; Bruland and
Richardson 2005; Moser et al., 2007) as it is simple and allows for straightforward
sampling designs, however, the visual characterization of hummocks and hollows is
subjective and has the potential to produce biased results for several reasons. First,
although microform vegetation and hydrology may be included in detailed study
site/method descriptions, these characteristics may be quite different for microforms
classified as hummocks at one study site compared to hummocks at a different study
site. Biogeochemical function (ecosystem fluxes) may differ for microforms within a site
(*e.g.* Bubier et al., 1993; Pelletier et al., 2011), but if the vegetation and hydrology of
those microforms vary for different peatlands, assumptions for hummock and hollow
biogeochemical function at one site may not be applicable to other peatlands. Given
that there may also be large differences in the relative/absolute height and surface
roughness of microforms between sites, comparing studies with hummock and hollow
microforms as a central component of the sampling design can be problematic.
Moreover, the surface area, spatial distribution, and relative proportion of hummock and



91 hollow microforms present within a peatland also vary between sites (*e.g.* Moore et al.,

92 2015), which may introduce bias into sampling design. For example, researchers may

93 over-sample the visually obvious extremes of the hummock-hollow continuum. Given

94 that several peatland hydrological and ecosystem carbon models parameterize peat

95 decomposition, production and hydraulic properties based on peatland microform

96 classification (*e.g.* Dimitrov et al., 2010; Sonnentag et al., 2008), the aforementioned

97 sampling and classification biases may also lead to issues in determining the scale and

98 complexity required for ecosystem modelling (*e.g.* Larsen et al., 2016).

99

100 The construction of a digital elevation model (DEM) in a peatland allows for the

101 classification of microforms based on quantitative measures (*e.g.* relative position, slope,

102 or roughness) (*e.g.* Mercer and Westbrook, 2016; Rahman et al., 2017) rather than

103 relying on qualitative/visual methods. Given the wide use and adoption of the hummock-

104 hollow conceptual framework, we examine the potential utility of DEM quantitative

105 techniques to overcome the concerns with the dominant qualitative hummock and

106 hollow framework/classification scheme. As such, the two main objectives of this study

107 were to: (i) provide a geostatistical/geospatial description of plot scale microtopographic

108 variation in peatlands; and (ii) to use simple physically-based and empirical models to

109 examine the effect of measured microtopographic complexity on ecosystem fluxes. For

110 the first objective, our two main focuses were: i) using a case-study approach, assess

111 how much area needs to be sampled in order to be able to adequately quantify

112 microtopographic variability within an unpatterned peatland; and ii) using multi-site plot-

113 scale sampling, explore DEM-derived morphometric properties (*e.g.* microtopography



height distribution, slope, and roughness) of peatland microforms which may be useful
as microtopographic metrics.

**METHODS**
*Experimental design*
We first evaluated how much sampling area is needed to capture the overall
microtopographic variation of an unpatterned site using both structure-from-motion (SfM)
(see Brown and Lowe 2005; Mercer and Westbrook 2016) and a transect based
sampling approach. To accomplish this, we randomly sampled 50 plots for SfM
reconstruction in a peatland near Red Earth Creek, AB (56.54°N 115.22°W) (hereafter
referred to as site-level). In addition, we manually measured surface elevation along
several 50 m transects at 0.05 m intervals covering the plot area at the Red Earth Creek
site. Secondly, we used SfM to examine morphometric properties at the plot scale in 9
boreal/hemi-boreal, non-permafrost, ombrotrophic peatlands (4 in Canada, 4 in USA, 1
in Sweden; see Table 1) using two different approaches. The first approach involved
randomly selecting 9 plot locations within a single site and creating a plot around the
random location which was perceived to contain a hummock-hollow pair. The second
approach involved qualitatively choosing what was perceived to be a representative
hummock-hollow pair at 9 different sites. The aim of our approach was to highlight the
potential breadth of variation in morphometric properties which might be observed either
within a site (*i.e.* implications for small sample size) or across sites (*i.e.* highlight
potential challenges with site inter-comparisons without supporting information of
peatland microtopographic metrics). For both randomly located plots and qualitatively



chosen plots, individuals were asked to identify a central point for a hummock and
hollow subplot within the larger microtopography plot.
***Site preparation and image acquisition protocol***
All vascular vegetation was removed from the plot area using scissors and hand
pruners in order to provide an unobstructed view of the surface microtopographic
variation (moss surface) for imaging. Matte-colored discs ($n$=20) of 0.04 m diameter
were placed randomly on the clipped surface to provide reference points for better
correlation between images. To provide absolute scale and orientation, two boxes of
known dimensions (0.1×0.1×0.1 m) were placed in each plot and levelled prior to image
acquisition. Images of each target area were taken via at least two circuits around the
plot,   with   images   taken   from   two   separate   vertical   viewing   angles   (see
http://www.cs.cmu.edu/~reconstruction/basic_workflow.html for third party description of
general workflow). Distance to target area was set so that a large portion of the clipped
area was visible in each image. To produce different horizontal viewing angles, images
were taken every one or two paces around the perimeter of the plot. This procedure
yielded 41 to 282 overlapping images from multiple view-points of the plot areas, which
ranged in size from 3.2 to 10.1 m$^2$ (Table 1). Images were taken during either clear-sky
or over-cast conditions near mid-day during the summer to avoid changing lighting
conditions and to limit self-shadowing of the surface. Images were captured with digital
cameras   using   automatic   exposure   settings.   Prior   to   analysis,   all   images   were
downscaled where necessary to a common resolution of 2048 x 1536 using a Lanczos3
filter.



### *Digital elevation models of microtopography*

A point-cloud of the moss surface was generated using an SfM approach (Brown and Lowe 2005; Mercer and Westbrook 2016) using the program Visual SfM (Wu, 2011). Visual SfM identifies image features for cross-comparison using a scale-invariant feature transform (Lowe, 1999), and then matches features between images in a pairwise manner. Effectively, this creates multiple stereo-pairs from which camera position and scene geometry can be estimated through triangulation. This procedure yielded average point cloud densities ranging from 3-59 pixels cm$^{-2}$ for the imaged plots (Table 1).

Prior to generating the DEMs, point clouds were cropped to the region of interest (*i.e.* area of clipped vegetation), then scaled, levelled, and oriented using the rendered reference objects. DEMs were produced using the MATLAB function *TriScatteredInterp* (MATLAB R2010a, The Mathworks), which performs Delaunay triangulation of the point clouds. DEMs were generated on a 0.01 x 0.01 m grid using natural neighbor (Voronoi) interpolation. The DEMs were smoothed using a mean filter window with a size of 0.03 x 0.03 m. Finally, a mask was applied to the DEMs to remove reference objects.

### *Capturing site-level microtopographic variation*

Plots from the Red Earth Creek peatland were ~3.5 m$^2$ and differences between plot elevation for the 50 plots were surveyed using a Smart Leveler digital water level (accuracy ±2.5 mm), with offsets applied to DEMs. A Monte Carlo re-sampling approach was used to evaluate how total variance in microtopographic elevation increased with



increasing sample size. For each sample size (*i.e.* 1-50), 200 random re-samplings
were performed. To estimate the change in variance with increasing sample size, a
rectangular hyperbola was fit to the mean variance (y) versus sample size (x):
$y = \dfrac{ax+b-\sqrt{(ax+b)^2-4axbc}}{2c}$
where *b* is the estimated maximum total variance, and *a* and *c* are initial slope and
concavity parameters.

To evaluate the dominant scale of microtopographic variation which contributes to total
variance, a fast Fourier transform (*fft* function in MATLAB) was used to estimate the
power spectral density (PSD) of microtopographic variation along an artificially
constructed 300 m long transect (combination of multiple transects). Manual
measurements of moss surface elevation were taken every 0.05 m along six 50 m
transects at the Red Earth Creek, AB and Nobel, ON site using the Smart Leveler.

***Plot-level microtopographic variation***
Plot-level microtopographic variation was analyzed using randomly and qualitatively
chosen plot locations listed in Table 1. Based on the hummock-hollow conceptual model,
our *a priori* assumption was that a hummock-hollow pair would have a bi-modal
distribution of surface elevation. Our null hypothesis was that microtopography would
follow a bi-modal distribution, so we evaluated DEM height distributions using 1– to 3–
member Gaussian mixture models (GMM) to evaluate whether 2-member GMMs would
best explain height distributions. GMMs were fit to DEM height distributions using the
MATLAB function *gmmdistribution.fit*, which uses an iterative expectation maximization



algorithm to determine GMM parameters representing maximum likelihood estimates.
The GMM fit function was seeded with initial parameter estimates using $k$–means
cluster analysis. The best model was decided based on the minimum Akaike
information criteria (AIC).

Surface slope and aspect were evaluated using the computed surface normals for each
point and eight connected neighbours of the DEM. The fractal dimension of plots was
evaluated using radially averaged PSD derived from an *fft* of elevation data. The Hurst
($H$) exponent (values of 0–1) presented herein is related to fractal dimension as 3-$H$,
where the slope of the PSD curve in log space is -2(H+1).

### *Modelled moss surface insolation and productivity*

Potential moss surface insolation was modelled using the formulation presented in
Kumar et al. (1997) to account for earth-sun geometry, surface slope and aspect, and
diffuse radiation under clear-sky conditions. Total potential insolation was evaluated on
an annual basis and normalized relative to total insolation on a flat surface for each plot
location.

For moss net primary productivity (NPP) and capitula water content ($WC$), each plot
was classified into three units based on relative elevation which notionally correspond
with hollow/lawn, low hummock, high hummock. K-means clustering was used to
perform unsupervised classification of microtopographic elevation (Figure S1). A
separate parameterization for moss NPP and WC was used for each elevation cluster.





Parameterizations for hollow/lawn, low hummock, and high hummock were obtained
from *Sphagnum* species of the section Cuspidata, Sphagnum, and Acutifolia,
respectively (Figure S2). An empirical relation between WC and water table depth (WTD)
was modelled as follows:
$$WC = p_1 \cdot \ln(p_2 \cdot WTD) + p_3$$
where WC is in $g_{water}$ $g^{-1}_{dry\ weight}$, and $p_{1-3}$ are fitted parameters. WC was restricted to a
range of 1–25 $g_{water}$ $g^{-1}_{dry\ weiight}$. A rational function was used to model the relation
between moss capitula NPP and WC:
$$NPP_{pot} = 100 \cdot \left( \frac{p_4 \cdot x^2 + p_5 \cdot x + p_6}{x^2 + p_7 \cdot x + p_8} \right) \cdot NPP_{max}^{-1}$$
where $NPP_{pot}$ represents % of maximum NPP, and $p_{4-8}$ are fitted parameters. Estimates
of 83, 170, and 198 g $m^{-2}$ $mo^{-1}$ for $NPP_{max}$ were used to represent *Sphagnum* species of
section Cuspidata, Sphagnum, and Acutifolia, respectively (Nungesser, 2003).


**RESULTS**
***Site-level microtopographic variation***
In characterizing microtopographic variability across the Red Earth Creek site, our data
shows that variability in surface elevation increases asymptotically with sample size (*i.e.*
area sampled) and is well predicted by a rectangular hyperbola ($r^2$=0.98; p<<0.01) (Fig.
1). Based on the asymptote of the fitted rectangular hyperbola (0.147 m), Figure 1
shows that on average an area of 32 $m^2$ (*i.e.* 9 random plots of ~3.5 $m^2$ size) contains
roughly 95% of the predicted site-scale microtopographic variability. Even though
increasing the number of plots by a factor of 5 (*i.e.* ~50 plots) has little effect on the



average variance in surface elevation, the range associated with re-sampling is reduced
by about half (Fig. 1 – shaded area).

While the Red Earth Creek multi-plot DEM data provides the ability to assess the area
required to capture site-scale microtopographic variability for a small unpatterned
Alberta peatland, it does not directly provide information on what spatial scales
contribute most to overall variability. The power spectral density (PSD) of manual
elevation transects from both the Red Earth Creek and Nobel sites suggests that most
of the microtopographic variation for these two surveyed sites occurs at spatial scales
between 1–10 m (Fig. 2 – cumulative curves). Both sites have qualitatively similar PSD
curves in log-space with a roll-off at spatial scales between 2.6–3.1 m (break point of
piecewise regression). Moreover, the PSD of microtopographic variation appears to be
well described by a power law (*i.e.* relatively smooth slope in log space) at small spatial
scales resulting in a Hurst exponent (see Methods for relation to fractal dimension)
between 0.61–0.79.

***Plot-level hypsometry and fractal dimension***
There is a characteristic difference in the elevation distribution of whole-plots compared
to that of the corresponding hummock-hollow subplots for both qualitatively (Fig. 3) and
randomly (Fig. 4) chosen plot locations. The elevation distributions for hummock-hollow
subplots tend to have a clear separation of modes (Fig. 3-4 B-panels). The degree of
separation in modes has a weak ($r^2$ = 0.31) but significant linear relation ($F_{16}$ = 7.1, $p$ =
0.017) with the microtopographic range in the whole plot. On average, the elevation



range absent from the hummock-hollow subplots represents roughly 25% of the
microtopographic range of the whole plot. When all hummock-hollow subplots are
aggregated across randomly selected plots (*i.e.* Nobel, ON site), the whole elevation
distribution is captured (Fig. S3). However, there remains a bias towards higher
elevations being sampled in the aggregated subplot elevation distribution compared to
the aggregated whole plot elevation distribution.

In testing the null hypothesis of bimodally distributed relative surface elevation at the
plot scale, we examined the goodness of fit of one-, two-, and three-member GMMs. An
assessment of all 18 plots suggests that two- or three-member GMMs tend to provide a
better fit to reconstructed elevation distributions compared to a one-member (*i.e.* normal)
distribution. Based on AIC values, the one-member GMM was best for only 3 plots,
while two- and three-member GMMs were best for 6 and 9 plots, respectively (Table 2).
In contrast, when GMMs were fit to hummock-hollow subplot data, the two-member
GMM tended to outperform one- and three-member GMMs.

The mean ($\mu$) and standard deviation of elevation for hummock and hollow subplots
were grouped and compared according to plot selection method (*i.e.* random within site
versus qualitative between site selection). Since the $\mu$ parameter corresponds with
relative elevation, we took the difference between the two members (*i.e.* $\mu_{hum}$–$\mu_{hol}$) for
comparison purposes. Overall, the qualitatively chosen plots appear to have similar
($F_{1,16}$=0.2; p=0.68) relative hummock heights ($\mu_{hum}$–$\mu_{hol}$) (0.21±0.08 m) compared to the
randomly chosen plots. (0.19±0.09 m). Variation in elevation tended to be lower in





hollow subplots (0.032±0.012 m) compared to hummock subplots (0.022±0.009 m)
(microform; $F_{1,32}$=9.0, p=0.005), where the difference between hummock and hollow
subplots was similar when comparing qualitatively and randomly chosen sites
(microform × plot type; $F_{1,32}$=0.02; p=0.89).

Depending on the underlying structure of spatial variability, surface roughness can be
highly dependent on the scale of analysis. A two-dimensional power spectral density of
elevation provides a means to formally describe the change in roughness with scale
(Fig. 5). The power spectral density of elevation was found to be a linear function of
length-scale across the 0.05–1 m range in log–log space ($r^2_{adj}$>0.96) and is the basis for
the Hurst exponent ($H$) (see methods for relation to fractal dimension). While the
distribution of $H$ for qualitatively chosen plots (0.73±0.18) was higher compared to
randomly chosen plots (0.60±0.11) (*i.e.* comparatively less 'complexity' at finer spatial
scales), the difference was not strongly significant ($F_{1,16} = 3.63$; $p = 0.075$).

***Plot-level slope, aspect and solar insolation***
A Weibull distribution provided a good fit to the slopes for the reconstructed DEMs (Fig.
S4), where the average, maximum, and minimum RMSE were 0.10%, 0.14%, and
0.06%, respectively, based on a relative frequency distribution with 1° bin sizes. When
grouped according to qualitatively versus randomly chosen plots, the modal slope for
whole plots was 21.5±4.4° and 23.4±5.7°, respectively. Similarly, the distribution of
standard deviation in slope for randomly and qualitatively chosen plots was 14.6±1.3°
and 14.5±2.1°, respectively. Comparing the parameter distributions from the Weibull fit





for qualitatively and randomly chosen plots, it was found that there was no significant
difference in the mean scale (analogous to mode) and shape (analogous to standard
deviation) parameters (scale: p=0.44, $F_{1,16}$=0.62; shape: p=0.88, $F_{1,16}$=0.02).

While modal slope tended to only be slightly higher in the hummock subplots
(22.9±6.8°) versus hollow subplots (19.5±6.0°), there was greater distinction in the
prevalence of steep slopes (*i.e.* >45°) in hummock subplots (14.8±10.4%) versus hollow
subplots (8.4±9.5%) (Fig. S5). Comparing slope in the hummock/hollow subplots to the
3-member GMM clusters (high, intermediate, and low elevations – for example see Fig.
S1), we see that the subplots tend to be somewhat flatter compared to the rest of the
plot, particularly for hollow subplots (Fig. S5).

Figure 7 shows how slope and aspect affect potential solar insolation at the moss
surface under ideal conditions (i.e. clear-sky, sparse vegetation). Potential solar
insolation is significantly affected by aspect ($F_{7,60820} \geq 290.8$, $p<<0.01$) and its interaction
with slope ($F_{7,45606} \geq 7043.7$, $p<<0.01$), where on average, south facing slopes receive
double the potential solar insolation compared to north facing slopes. Based on
measured slope and aspect at randomly and qualitatively chosen plots, median
potential solar insolation for a south aspect is 12-24% greater compared to a flat
surface. Similarly, for a north-facing aspect, median potential solar insolation is 18-40%
lower (Figure S6).

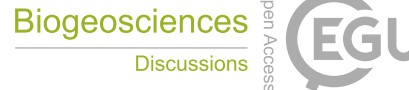

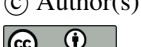

### *Plot-level empirical model of moss productivity using high resolution DEMs*

Assuming a flat water table at the plot-level, Figure 8 shows how modelled $NPP_{pot}$ varies with WTD relative to the average hollow surface. Hollows tend to have a comparatively narrow range of WTD (i.e. 0–0.15 m) over which the moss is expected to be highly productive compared to hummocks. Despite using species-dependent $NPP_{pot}$-WC relations, the large differences in water table range over which hummock and hollow $NPP_{pot}$ is high is largely driven by the WC-WTD relations (Figure S2). Where moss species have large differences in $NPP_{max}$ and different characteristic water retention, $NPP_{pot}$ rarely overlaps between microtopographic classes (Figure 8). If we ignore the effect of species-dependent characteristics (i.e. $NPP_{max}$, $NPP_{pot}$–WC, and WC–WTD) and use a single average parameterization, differences between microtopographic classes tend to be smaller for shallow water table conditions (Figure S7), yet there remains a characteristic difference in mean $NPP_{pot}$ between microtopographic classes.

From a scaling perspective, modelled $NPP_{pot}$ (Figure 8 and S7) were used to compare spatially explicit estimates with plot averages based on the notional chamber subplot (i.e. pre-determined 0.37 $m^2$ area in perceived hummock and hollow — see methods). In general, spatially explicit $NPP_{pot}$ estimates tended to be higher/lower than the hummock-hollow estimates depending on whether the water table was relatively shallow/deep (Figure 9a). The maximum positive bias between the spatially explicit and hummock–hollow $NPP_{pot}$ values ranged from 21.1–40.1 g $m^{-2}$ $mo^{-1}$, while the negative bias ranged from -5.9 to -40.9 g $m^{-2}$ $mo^{-1}$. Using a single average parameterization for



NPP$_{pot}$ tends to result overwhelmingly in positive bias between the spatially explicit and
hummock-hollow models, where maximum bias ranges from 22.7 to 58.9 g m$^{-2}$ mo$^{-1}$.
Averaged across all 18 plots, the subjective hummock subplot broadly overlapped with
the k-means high-hummock classification (94%), with only small portions overlapping
with the low-hummock classification (6%). Similarly, the subjective hollow subplot
broadly overlapped with the k-means hollow/lawn classification (79%), with only small
portions overlapping with the low-hummock classification (20%). In this study, our
results indicate that the subjective choice of hummock and hollow subplot location (e.g.
for chamber flux measurement) systematically under samples intermediate topographic
positions. This is exemplified in Figure S8 which shows the spatial distribution of NPP$_{pot}$
for one of the plots. For the NPP$_{pot}$ model using separate parameterization for the
microtopography classes, the low-hummock class remains distinct from both the
hollow/lawn and high-hummock class except under very dry conditions. For the uniform
parameterization, the low-hummock classification is distinct from the other two classes
under wet conditions, behaves like the hollow/lawn under moderately dry conditions,
and behaves like a hummock under very dry conditions.


**DISCUSSION**
***Assessing microform representativeness***
In studies which use the hummock-hollow microtopography classification as part of their
sampling design, there are many cases in which the plot choice is said to be
representative (*e.g.* Kettridge and Baird 2008; Laing et al., 2008; Nijp et al., 2014), but



often lacks detail on how representativeness was assessed. For example, when
characterizing the surface within an eddy covariance flux measurement footprint, it is
common to only sample one or few hummock-hollow pair(s) (*e.g.* Lafleur et al., 2003;
Humphreys et al., 2006; Peichl et al., 2014; Moore et al., 2015). Similarly, for direct
measurements of surface fluxes where microtopography is considered explicitly,
chamber-based measurements typically use between four and eight replicates (*e.g.*
Frenzel and Karofeld 2000; Turetsky et al., 2002; Forbrich et al., 2011; Petrone et al.,
2011) per microtopographic unit. For peatland studies which use random plots, as many
as 30 plots per site have been reported (*i.e.* Wieder et al., 2009), yet earlier studies
have reported using as few as one to four plots to characterize a site (*e.g.* Crill et al.,
1988; Shannon and White 1994; Regina et al., 1996). Using the Red Earth Creek
results as a reference, for studies which have 4-8 replicates, 2-3 microtopographic units
(*e.g.* hummock, lawn, hollow), and the more common chamber size of roughly 0.6 x 0.6
m, we would infer from our results that the typical total sample area for chamber flux
measurements in a peatland ecosystem would capture on the order of 70-86% of site-
scale microtopographic variability in their plots. It should be noted however that the
simple assessment above assumes that chamber placement is random. In cases with
lower replication of two microtopographic units, our results suggest that the uncertainty
associated with repeated sampling is relatively high (Fig. 1 – shaded area) and that the
choice of two microtopographic units could lead to an under-sampling of intermediate
topographic positions (*e.g.* Fig. 3-4 B-panels). When the ecosystem processes of
interest are not measured across the range of variability observed at the site-scale,
particularly for non-linear processes, then scaling from process-based, or simply plot-





scale measurements, is at risk of being biased. Our simple empirical model of moss
$NPP_{pot}$ demonstrates that flux bias can be large relative to $NPP_{max}$ and is strongly
dependent on water table depth (Figure 9). Although $NPP_{pot}$ estimates are strongly
influenced by the parameterization used (e.g. Figure 8 and S7), there remains a large
bias between the spatially explicit and hummock-hollow $NPP_{pot}$ models.

To upscale models or plot-scale measurements it is important to determine the
microtopographic structure and variability of a peatland. There were often non-bimodal
distributions of microtopography in our study sites (Fig. 3–4 A-panels and Table 2)
where the more continuous distribution of elevation at the plot scale suggests that when
experimental designs use hummock-hollow pairs as the primary experimental unit (Fig.
3–4 B-panels) they have a tendency to capture the ends of the distribution, omitting on
average 25% of the elevation distribution at the plot scale (see also Figure S3). In this
study we clipped vegetation in 50 small random plots to produce very high resolution
DEMs for assessing microtope-scale (*i.e.* S3 hummock-hollow complex, *cf.* Belyea and
Baird, 2006) variability, yet surface vegetation removal will generally be undesirable.
Ground- or drone-based SfM approaches have been used to produce a digital surface
model (DSM – vegetation present) for alpine (Mercer and Westbrook, 2016) and blanket
(Harris and Baird, 2018) peatlands with reasonable accuracy (e.g. mean absolute error
of ~0.08 m, and normalized median absolute deviation of ~0.11 m for the alpine and
blanket peatlands, respectively). In situations where surface vegetation removal is not
possible or desirable and/or where drone-based imagery is hampered (*e.g.* treed
peatlands), a survey of height distribution along one or several transects would provide



an alternative to assessing microtope to mesotope-scale (S3–S4 Belyea and Baird,
2006) microtopographic variability. The power spectral density of transect data would
suggest that, for absolute height, a sampling interval of less than 1 m (*e.g.* 0.5 m) for
several 50 m transects would capture the scales of variability which contribute most to
total height variance (Fig. 2 and 5) since this corresponds to ~90% of measured
microtopographic variation and provide sufficient fine-scale data to estimate the fractal
dimension of microtopography. Information on height distributions could provide the
basis for plot selection, where plots could be chosen to deliberately span the range of
variability, or to avoid oversampling extremes. Information on the height distribution
would furthermore provide the ability to scale up findings from the plot level given their
relative position in the wider distribution of microtopographic variability (*cf. Griffis* et al.,

446    2000).


Despite the variety of site characteristics observed, our plots were limited to bogs and
poor fens, and did not include sites with ridge and pool patterning. Nevertheless, our
results would suggest that generalizations based on a hummock-hollow classification,
either to the site-scale, or to hummocks-hollow pairs across sites should be viewed with
a degree of skepticism when sample size is low, or when a general microtopographic
survey is absent/unreported. Thus, for wider inter-comparability of peatland studies, SfM
or transect-based approaches of measuring and reporting on one or several
morphometric properties of microtopography could provide a more comprehensive
dataset to aid in future meta-analysis/synthesis.



*Implications for appropriate complexity ecosystem modelling in peatlands*

The complex shape/structure of peatland microtopography has generally been ignored from a modelling standpoint, but several studies have shown, for example, that slope and aspect may affect peat temperature (Kettridge and Baird 2010). Under clear-sky conditions, modelled annual total solar insolation differs from a flat surface by roughly ±20% in our measured plots, where our study sites span 43° to 60°N latitude (Figure S6). For north and south facing slopes, this effect is amplified (Figure 7) particularly for high- and low-hummock microtopographic classes (e.g. Figure S1) which tend to have greater average slope compared to the hollow/lawn classification (Figure S5). While our study sites are limited to the non-permafrost boreal region, the applicability of slope and aspect considerations to modelling tundra tussocks in arctic and permafrost regions is also relevant (*e.g.* De Baets et al., 2016). Based on the results of empirical studies, the shape of microtopographic features aught to play a role in ecosystem fluxes due to the effect of shortwave radiation on surface evaporation (Kettridge and Baird, 2010), photosynthetically active radiation on moss production (Harley et al., 1989; Loisel et al., 2012), and soil temperature on methane production and respiration (*e.g.* Lafleur et al., 2005; Waddington et al., 2009). It is important to note, however, that under cloudy conditions the increasing proportion of total insolation from diffuse radiation decreases the disparity in insolation associated with slope and aspect. Furthermore, in peatlands where substantial tree, shrub, or graminoid cover exists, the importance of slope and aspect on soil heating or ecosystem fluxes is likely to be low since insolation decreases exponentially with increasing vascular leaf area.





In addition to microtopographic shape/structure, the size of microtopographic features
and their small-scale variability can similarly affect ecosystem fluxes, where height
above water table imposes a first order control on water availability. Methane fluxes
from peatlands, for example, have been shown to vary logarithmically over 0.1 m scales
(Turetsky, 2014). Water availability at the moss surface has been shown to be both
species-dependent and strongly affected by water table (Hayward and Clymo, 1982;
Rydin, 1985), where moss species and water availability has been linked to many
ecohydrological processes such as surface evaporation (Kettridge and Waddington,
2014), productivity (Williams and Flanagan, 1998; Strack and Price, 2009), and
hydrophobicity (Moore et al., 2017). We show that when microtopographic variability is
explicitly modelled, complex patterns of potential moss productivity emerge (Figure S8)
which are not captured by a hummock-hollow model (Figure 9), and that the presence
of bias is independent of whether moss species niche partitioning is considered.

The SfM method is a potentially useful tool for examining both how morphometric
properties of the surface which affect ecohydrological processes vary within a site.
Moreover, information on microtopographic variability and structure from SfM-derived
DEMs can be used to further examine the potential role of fine-scale microtopographic
variability on biogeochemical processes within a modelling domain. The GMM is a
simple way to include a more realistic description of height distributions within
distributed peatland models (*e.g.* Dimitrov et al., 2010), or extend from the meso- to
micro-scale (Sonnentag et al., 2008). Computationally, GMMs are a relatively efficient
way of representing microtopographic variability, needing only two parameters per



member of the GMM distribution. Conceptually, the GMM distribution can be applied
directly in distributed peatland models to populate relative heights of individual cells. In
the case of one-dimensional models, a GMM distribution can be used as a transfer
function for any water table dependent processes, particularly in cases where the
relation is non-linear. Alternatively, a small number of parameters from the PSD of
microtopographic elevation (*e.g.* variance, Hurst exponent, and spatial scale of break
point), be it from a DEM (Fig. S4) or transect (Fig. 2), can be used to generate 'synthetic'
microtopography which includes spatial structure in elevation change rather than just
the distribution.

**CONCLUSIONS**
The magnitude of variation in assessed morphometric properties within a site (randomly
chosen plots) is commensurate with the range across sites (qualitative plots) where
mean differences are comparatively small. With a small effect size, our results highlight
the need for adequate spatial sampling in process-based studies of microform function,
particularly when upscaling to the whole peatland or in order to make broader
inferences regarding peatland microforms in general. The SfM technique provides very
high resolution and accurate DEMs relatively quickly and easily. For studies which focus
on processes which are correlated with microtopographic position, a DEM or DSM
derived from ground- or drone-based imagery provides valuable information on
microtopographic variability and structure which can help inform plot selection, be used
for upscaling results, and quantify well defined morphometric and topographic variables
to aid in study inter-comparisons. Conversely, height measurements (*e.g.* using a dGPS



or other survey method) along a transect of at least 100 m with measurements taken at
an interval of less than 1 m provides sufficient information to describe a number of
peatland morphometric properties (*e.g.* hypsometry, roughness, fractal dimension, etc.).

Our study highlights the need to critically assess sampling approaches in peatland
ecosystem science where we show that a strict hummock-hollow classification tends to
under-sample intermediate topographic positions. While the discretization of peatland
ecosystems into microtopographic units has facilitated the understanding of peatland
processes in the context of species niche partitioning and their covariates such as water
table position, we now have techniques to better quantify variability with relative ease.
Consequently, techniques such as SfM enable us to consider peatland ecosystem
processes as part of a continuum. We must recognize that our conceptualizations, while
perhaps representing necessary simplifications, ought to be scrutinized to ensure that
elements of peatland complexity are not omitted. By considering microtopography
explicitly, we may be better able to understand how ecosystem complexity subsumed
within current microtopographic classifications might represent an important
unquantified confounding variable which limits our ability to adequately resolve and thus
understand certain peatland processes.

**DATA AVAILABILITY**
The post-processed point clouds used to generate digital elevation models which were
analysed in this study are available online at: [File are currently uploaded to a project



folder on Zenodo. Final publishing and assignment of DOI will be completed after review,
where additional material may be added based on recommendation(s) from reviewers].

**ACKNOWLEDGEMENTS**
We would like to thank James Sherwood and Paul Morris for valuable conversations
regarding the feasibility of this study and early discussions regarding research design.
We thank Lorna Harris for comments on an earlier draft of this manuscript. We also
thank Tom Ulanowski for data collection for the James Bay site, Rebekah Ingram and
Kristyn Mayner for data collection at the Red Earth Creek site, Mandy MacDougall,
Alanna Smolarz and Alex Furukawa for assistance with the Nobel data collection and
analysis, and to Lee Slater for data collection in Maine. This research was supported by
a NSERC Discovery Grant and NSERC Discovery Accelerator Supplement to JMW.



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



**Table 1: Summary information on sample locations and SfM reconstructions of**
**microtopographic variation for target areas for randomly and qualitatively chosen**
**plot locations within a site.**

| Location | Plot Name | Lat. (°N) | Lon. (°W) | Plot Area (m²) | Number of Images Used | Point Cloud Density (m⁻²) |
|---|---|---|---|---|---|---|
| *Random* | | | | | | |
| Nobel, ON[1] | Alpha | 45.434 | 80.081 | 4.6 | 47 | $6.04 \times 10^4$ |
| -- | Beta | -- | -- | 3.8 | 41 | $7.83 \times 10^4$ |
| -- | Gamma | -- | -- | 4.1 | 44 | $6.68 \times 10^4$ |
| -- | Epsilon | -- | -- | 5.2 | 53 | $8.38 \times 10^4$ |
| -- | Zeta | -- | -- | 6.12 | 66 | $1.60 \times 10^5$ |
| -- | Eta | -- | -- | 5.74 | 60 | $1.42 \times 10^5$ |
| -- | Iota | -- | -- | 5.66 | 49 | $3.23 \times 10^4$ |
| -- | Kappa | -- | -- | 5.53 | 66 | $1.77 \times 10^5$ |
| -- | Theta | -- | -- | 5.48 | 59 | $1.38 \times 10^5$ |
| -- | Lambda | -- | -- | 8.2 | 61 | $1.18 \times 10^5$ |
| *Qualitative* | | | | | | |
| Caribou Bog, MN[2] | Maine | 44.83 | 68.75 | 10.1 | 79 | $3.75 \times 10^4$ |
| James Bay, ON[3] | JamesBay | 52.846 | 83.930 | 7.6 | 82 | $1.97 \times 10^5$ |
| Ottawa, ON | Limerick | 44.877 | 75.609 | 9.0 | 282 | $5.94 \times 10^5$ |
| Puslinch, ON[4] | Puslinch | 43.407 | 80.264 | 6.45 | 109 | $1.12 \times 10^5$ |
| Rödmossen, SWE[5] | Sweden | 60.013 | -17.355 | 10.6 | 105 | $4.71 \times 10^4$ |
| Seney, MI[6] | WET | 46.190 | 86.019 | 7.7 | 135 | $1.12 \times 10^5$ |
| Seney, MI[6] | INT | 46.192 | 86.019 | 7.0 | 109 | $9.44 \times 10^4$ |
| Seney, MI[6] | DRY | 46.186 | 86.015 | 7.3 | 62 | $8.89 \times 10^4$ |
| Nobel, ON[1] | Lambda | 45.434 | 80.081 | 8.2 | 61 | $1.18 \times 10^4$ |

For detailed site information see the following studies: 1. Moore et al., (2019); 2.
Kettridge et al. (2008); 3. Ulanowski and Branfireuen (2013); 4. Campbell et al. (1997);
5. Granath et al. (2009); 6. Moore et al. (2015).



**Table 2: Estimated parameters for one-, two-, or three-member Gaussian mixture model (GMM) fit to DEM**
**elevations. Results are presented for the GMM which minimizes AIC. Plots are separated into those chosen at**
**random versus qualitatively at their respective site.**

| Location | Plot Name | 1st distribution | | | 2nd distribution | | | 3rd distribution | | |
|---|---|---|---|---|---|---|---|---|---|---|
| | | Mean | SD | Scale | Mean | SD | Scale | Mean | SD | Scale |
| *Random* | | | | | | | | | | |
| Nobel, ON | Alpha | 0.11 | 0.03 | 0.23 | 0.20 | 0.03 | 0.36 | 0.28 | 0.06 | 0.41 |
| -- | Beta | 0.13 | 0.04 | 0.37 | 0.18 | 0.03 | 0.53 | 0.29 | 0.04 | 0.10 |
| -- | Epsilon | 0.07 | 0.02 | 0.06 | 0.18 | 0.05 | 0.30 | 0.31 | 0.05 | 0.64 |
| -- | Gamma | 0.19 | 0.08 | 0.23 | 0.26 | 0.04 | 0.59 | 0.44 | 0.06 | 0.18 |
| -- | Zeta | 0.11 | 0.03 | 1 | — | — | — | — | — | — |
| -- | Eta | 0.13 | 0.04 | 0.82 | 0.25 | 0.05 | 0.18 | — | — | — |
| -- | Iota | 0.11 | 0.03 | 0.24 | 0.19 | 0.06 | 0.76 | — | — | — |
| -- | Kappa | 0.11 | 0.04 | 0.23 | 0.23 | 0.06 | 0.60 | 0.42 | 0.05 | 0.06 |
| -- | Theta | 0.16 | 0.03 | 0.84 | 0.25 | 0.04 | 0.16 | — | — | — |
| *Qualitative* | | | | | | | | | | |
| Caribou Bog, ME | Maine | 0.07 | 0.02 | 0.15 | 0.16 | 0.02 | 0.55 | 0.28 | 0.07 | 0.30 |
| James Bay, ON | JamesBay | 0.17 | 0.08 | 1 | — | — | — | — | — | — |
| Ottawa, ON | Limerick | 0.08 | 0.02 | 0.38 | 0.15 | 0.05 | 0.62 | | | |
| Puslinch, ON | Puslinch | 0.14 | 0.053 | 1 | — | — | — | — | — | — |
| Rödmossen | Sweden | 0.17 | 0.05 | 0.87 | 0.36 | 0.04 | 0.13 | — | — | — |
| Seney, MI | WET | 0.23 | 0.08 | 0.59 | 0.36 | 0.05 | 0.25 | 0.44 | 0.03 | 0.16 |
| Seney, MI | INT | 0.25 | 0.07 | 0.51 | 0.45 | 0.06 | 0.40 | 0.53 | 0.02 | 0.09 |
| Seney, MI | DRY | 0.08 | 0.03 | 0.05 | 0.21 | 0.04 | 0.45 | 0.34 | 0.05 | 0.50 |
| Nobel, ON | Lambda | 0.05 | 0.02 | 0.46 | 0.20 | 0.08 | 0.54 | — | — | — |





**LIST OF FIGURES:**

Figure 1: Relation between standard deviation of microtopographic variation based on total sample area for the Red Earth Creek site based on fifty ~3.5 m$^2$ plots. The grey shaded area represents the 2.5 and 97.5 percentile of standard deviation from the Monte Carlo resampling procedure.

Figure 2: Absolute (solid lines) and cumulative (dashed lines) power spectral density of height along a 300 m transect for the Red Earth Creek, AB (red) and Nobel, ON (black) sites.

Figure 3: Relative frequency distribution of height in plots where a perceived representative hummock and adjacent hollow was subjectively chosen for a given site. Relative height distributions are shown for the entire plot (A) and for a hummock and hollow subplot (B) whose area corresponds to the size of a large flux measurement chamber. Elevations are referenced to the lowest point of the reconstructed surface and set to zero.

Figure 4: Relative frequency distribution of height in plots with randomly chosen locations within a site containing a perceived hummock and adjacent hollow. Relative height distributions are shown for the entire plot (A) and for a hummock and hollow subplot (B) whose area corresponds to the size of a large flux measurement chamber. Elevations are referenced to the lowest point of the reconstructed surface and set to zero.






Figure 5: Radially averaged power spectral density for randomly– (left panel) and
qualitatively– (right panel) chosen plots representing the change in elevation variability
with length scale. The slope between the power spectral density and wavevector
($2 \times \pi$/wavelength) in log-log space corresponds with the Hurst exponent ($H$), where
slope = $-2(H+1)$; and is related to the fractal dimension as $3 - H$.

Figure 6: Weibull probability density function of slope derived from surface normal of a
planar fit to elevation in a moving 0.03 m x 0.03 m window for all DEMs. Panels (a) and
(b) separate the randomly and qualitatively chosen plots, respectively.

Figure 7: Variation in potential solar insolation relative to a flat surface based on aspect
(a) and slope (b). Boxplots shows median and inter-quartile range, with outliers shown
as dots. Insolation as a function of slope has been bin averaged per cardinal direction,
where each point represents 100 data points. Slope and aspect data are for the Seney,
WET plot.

Figure 8: Mean potential net photosynthesis (NPP) for three microtopographic classes
(i.e. high-hummock, low-hummock, and lawn/hollow — see supplementary figure 1)
derived from spatially explicit elevation data for random (a,c) and qualitatively chosen
(b,d) plots. NPP-WC and WC-WTD relations are based on separate parameterization
for each microtopography class (see supplementary figure 2).


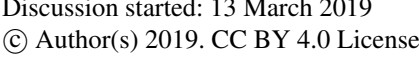
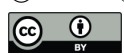

Figure 9: Difference in maximum potential net photosynthesis ($NPP_{pot}$) between models
using the measured distribution of elevation over the entire SfM-derived DEM and the
measured distribution within hummock-hollow subplots. $NPP_{pot}$ is modelled using
separate parameterization (Figure S2) for each microtopography class (a), as well as a
uniform (low-hummock) parameterization across microtopography classes (b).





[Figure 1]

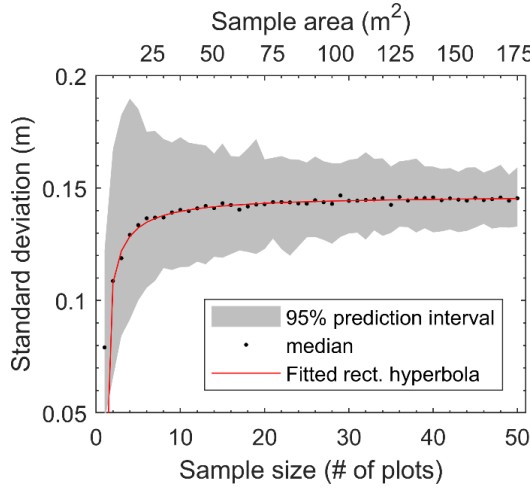





[Figure 2]

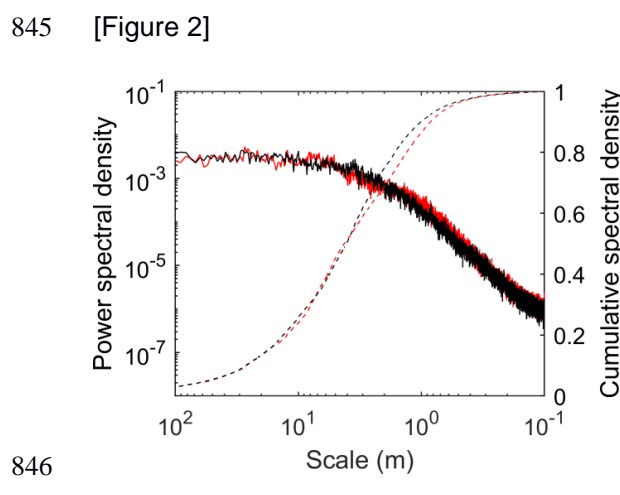





[Figure 3]

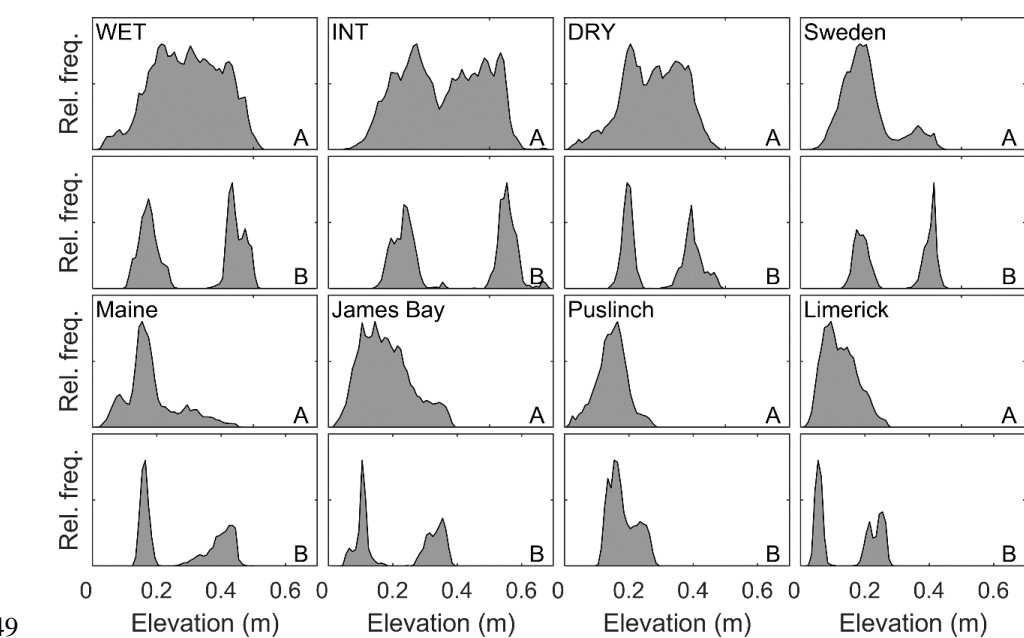







[Figure 4]

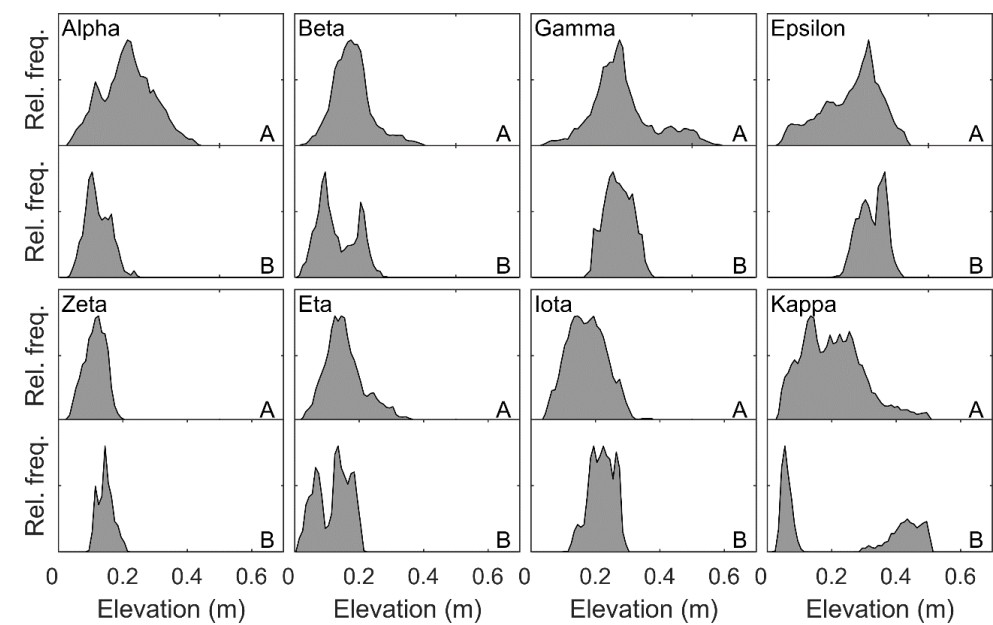







[Figure 5]

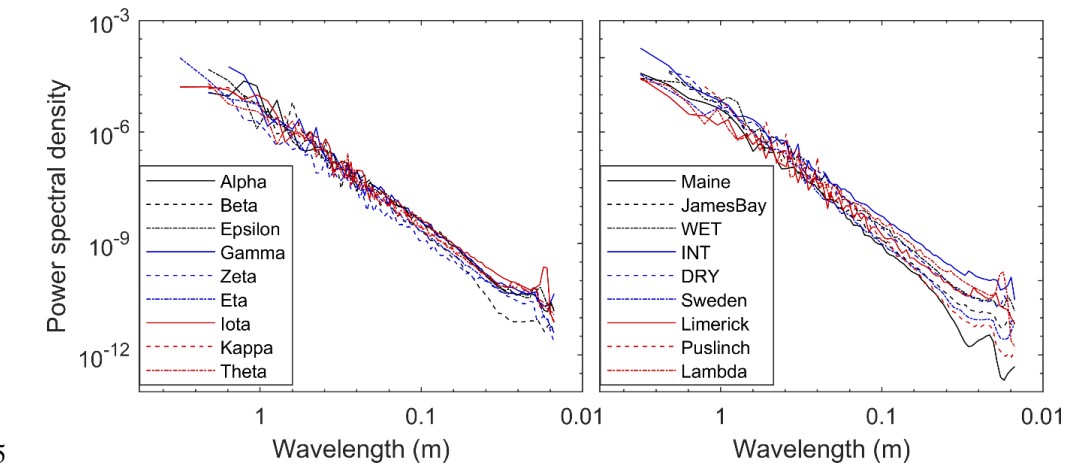







[Figure 6]

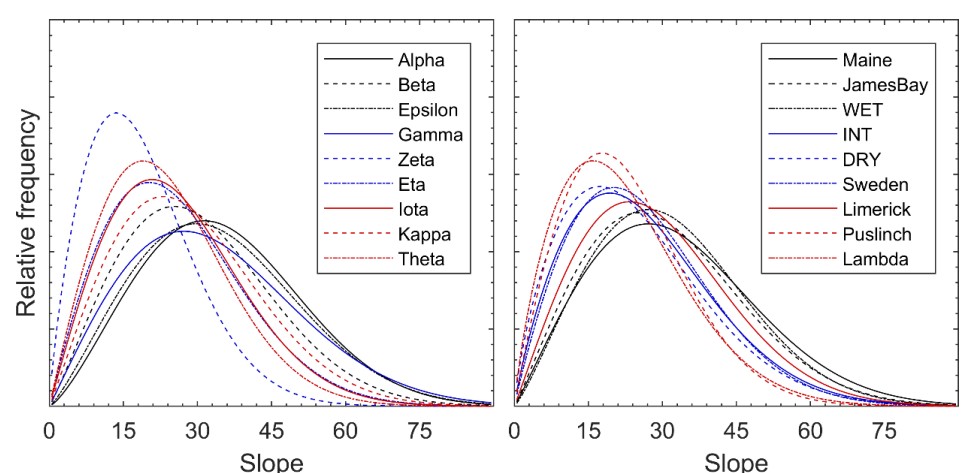







[Figure 7]

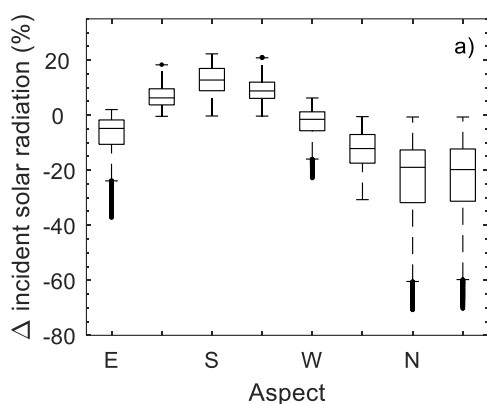
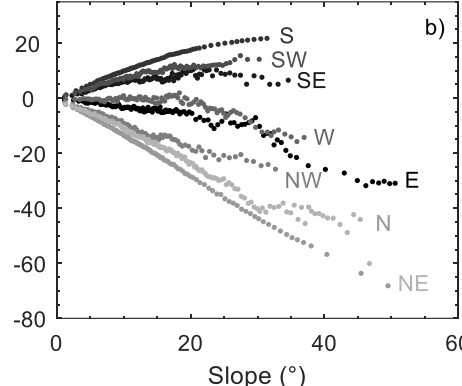







[Figure 8]

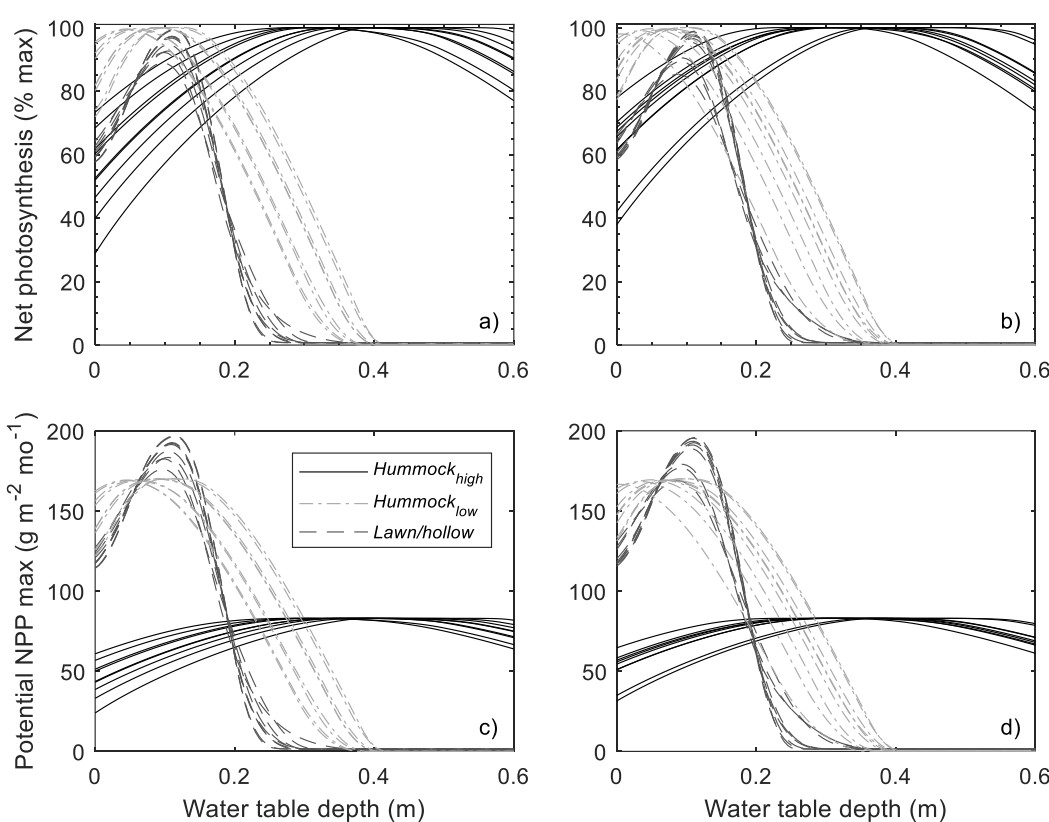





[Figure 9]

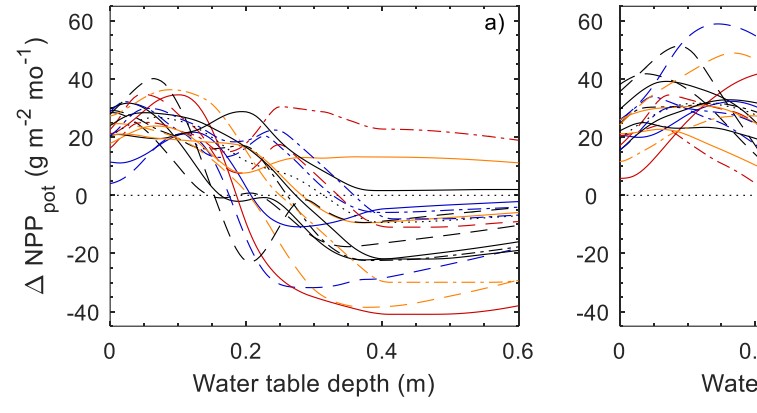
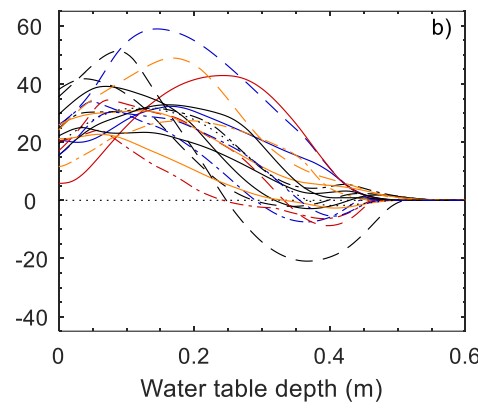

