# Peer review of "ASSESSING THE PEATLAND HUMMOCK-HOLLOW CLASSIFICATION"

_Biogeosciences, 2019_

## Referee Comment (RC1) · Anonymous Referee #1 · 2 Apr 2019

General comments

Moore et al. present a methodology to classify the peatland hummock-hollow variability for carbon flux modeling using high-resolution elevation models with k-means clustering. The study has collected samples across a variety of sites and mapped the high-resolution microtopography with the structure-from-model. This manuscript provides insights into the influence of microtopography on the uncertainties of field sampling and carbon flux modeling. Considering the importance of the peatland carbon fluxes to the global climate change, this study is relevant and necessary. Overall, this

manuscript is well-written and easy to follow. However, several issues in the manuscript need to be improved before the publication. This manuscript has done a nice analysis of the DEMs and evaluated its impacts on the NPP simulation. However, the validation of the generated DEM and the model-based fluxes is weak and needs to be strengthened. Another key issue for small scale flux simulation is to identify the optimal spatial resolution for modeling. I would also suggest the authors improve the analysis to identify the optimal spatial resolution to represent the microtopography. Generally, I think that this manuscript is publishable after revisions.

Specific comments

1. L20: Some key words are repeating from the title. Normally key words should be different from words in the title, as to provide additional information.

2. Abstract: The findings on the optimal spatial resolution is quite important for the appropriate complexity of flux modeling. As mentioned in L438-439, this manuscript concluded that on the optimal resolution to represent the spatial variability has been identified from Fig. 2 and 5. These findings should be reflected in the Abstract.

3. Validation of the generated elevation model is needed. The structure-from-motion technique is sensitive to the camera geometric calibration, camera position information, and the accuracy and numbers of ground control points. Validation results on the generated DEM are necessary.

4. L237: Equations should be marked with a number. For the equation at L237, the variable x should be explained.

5. L438-439: From Figure 5, it is not as easy as Figure 2 to identify the optimal resolution to represent the spatial variability. The authors can use additionally spatial analysis, e.g. semivariogram analysis, to strength these findings.

6. The structure-from-motion can provide both DEMs and orthophotos for the study site. In the manuscript, the authors have used the DEMs for data analysis. Potentially,

the orthophotos can be utilized to calculate vegetation indices to infer the vegetation growth conditions. This can improve the classification and NPP modeling. Why don't you make the best use of your data?

7. Table 1: Mistakes on the Longitude. For instance, Sweden should not be -17W. I guess the fourth column should be the elevation above the mean sea level instead of the longitude.

8. Table 2: In Table 2, some plots have been classified into three members. However, we cannot see such three members in the histogram distribution of Figure 3 and 4. Please explain the reason.

9. Figure S4. The curve should be from modeling and the dots are the measured one.

10. Figure S8: the scale bar should be added into the spatial map. Otherwise, readers don't know the spatial scale of these maps.

11. The paper has done a nice analysis of the carbon flux modeling and assessed the impact of water table depth on the carbon fluxes. However, it is hard to evaluate whether these modeling is accurate enough or not. It would be better to add some chamber or eddy covariance measurements to validate the simulated NPP. Or at least compare the results with other relevant studies.

---

## Referee Comment (RC2) · L. Kutzbach (Referee) · 27 May 2019

The manuscript of Moore et al. presents a very interesting and comprehensive analysis of the microtopographic structure of boreal non-patterned bogs. The paper scrutinizes the binary hummock-or-hollow classification approach, which is often followed in sampling design or modeling of biogeochemical and ecophysiological peatland processes.

The authors applied a well-designed combination of elaborate field data acquisition methods, targeted statistical analyses and appropriate process modeling. I am particularly pleased about the creative and thorough usage of various spatial statistical methods for analyzing the heterogeneity of peatland microtopography (e.g., Gaussian mixture models, Fourier transform power spectra of microtopographic variability along transects, slope and aspect analysis for microtopographic features, fractal dimension of plots). I also like the approach of simulating water content and net primary productivity in dependence of microtopography properties as an approach to demonstrate the relevance of thorough microtopography characterization for quantification of energy and matter fluxes. The authors show that non-consideration of the full continuum of microtopographical variability can lead to serious biases in spatial averages of net primary productivity due to negligence of microforms that are intermediate between hummocks and hollows. Even more pronounced bias would be expected for, e.g., methane emissions, which are controlled by water level depth below the moss surface in a highly nonlinear way.

Thus, the presented study is of high scientific relevance and originality. However, I think that the quality of the manuscript needs to be improved. In the following, I provide lists of (1.) general comments, (2.) specific comments, and (3.) technical comments. I recommend the manuscript for publication after major revisions.

General comments

(1) The experimental design of the study needs to be better explained. It is now too difficult for the reader to find out which method was applied where. That the many analyses were conducted at various peatland sites, needs to be more clearly stated already in the introduction. Furthermore, I think that a figure explaining the study design by including maps of different scale (e.g., northern hemisphere with location of all investigated peatlands, Nobel peatland with location of random plots in detail), would help. It would be also helpful if information on site and/or spatial scale would be added to all of the figure captions.

(2) The approach for modeling water content and potential NPP needs to be better

described (L. 224-240). What is the basis for the parameterizations for water content for the different microforms? Please provide references. Is NPP considered as a $CO_2$ flux or a carbon flux? Without specifying this, the modelled NPP values cannot be checked for plausibility. However, such a plausibility check would be necessary. Please compare your modelling results with empirical data on NPP of bog microforms.

Specific comments

L. 50: I do not like this often used comparison because it is like comparing apples with oranges: The carbon pool of peatlands is estimated over their mean peat depth (can be more than 15 m), whereas carbon pools of soils are estimated for specific reference soil depths (e.g. 1 m , 3 m). Hence, do peatlands contain one third of the upper meter of global soils or of the upper 3 m or how many meters? Furthermore, soils store not only organic carbon but also inorganic carbon!

L. 69: I would think that the area covered by a hummock can be also quite larger than 1 m2.

L. 96: I suggest adding the reference: Cresto Aleina F., Runkle B. R. K., Kleinen T., Kutzbach L., Schneider J., Brovkin V. (2015): Modeling micro-topographic controls on boreal peatland hydrology and methane fluxes. Biogeosciences 12: 5689-5704.

L. 112-113: Sentence not clear to me; please rewrite! I do not understand how you want to "explore DEM-derived properties" "using multi-site plot-scale sampling".

L. 137: Write more specific: What kind of "individuals"? Have these been scientists, students, or farmers neighboring the peatland?

L. 157: Unit of resolution?

L. 234: According to Si system, do not mix units and quantities. Better "WC is the ratio of the mass of water and the mass of the non-water components of the soil (Unit: g g-1)."

L237: Specify the variable x. Probably, x equals WC, correct?

L 238: Better: ". . .represents percentage of maximum NPP"

L. 836: It is confusing to use the two terms "net photosynthesis" and "NPP" as y-axis titles of different diagrams in the same figure, respectively. Do you use the terms as synonyms? In my view, integration of net photosynthesis over time at the canopy scale leads to NPP; thus "net photosynthesis" and "NPP" would be closely related but not synonymous.

Technical comments

L. 29: Correct "examine"

L. 31: Correct: "northern"

L. 38: Correct: "positions"

L. 50 Correct "one third"

L. 107: Hyphenate: "plot-scale"

L. 121: Hyphenate "transect-based"

L. 145: I suggest writing: " 0.1 m x 0.1 m x 0.1 m (same for similar expressions throughout the manuscript)

L. 179: Comma before "and" (beginning of independent sentence)

L. 186: Number the equations.

L. 208: better "selected" instead of "decided"

L. 239: "mo" is not a standard abbreviation for a SI unit. Please define this somewhere before using it.

L. 296: I would move the F statistics in parentheses to the end of the sentence.

L. 311: Infelicitous usage of statistical terminology: In my view, a result can be either significant or non-significant, give a specific error probability. It cannot be strongly of weakly significant.

L. 374: Hyphenate: "under-samples"

L. 380: Better a full stop instead of a comma after "conditions"

L. 465: Comma before "which"

L. 507: Hyphenate "water table-dependent"

L. 516: Comma before "where"

L. 532: Comma before "where"

---

## Author Comment (AC1) · 18 Jun 2019

**Reply to referee comments**

Manuscript:  ASSESSING THE PEATLAND HUMMOCK-HOLLOW CLASSIFICATION FRAMEWORK USING HIGH-RESOLUTION ELEVATION MODELS: IMPLICATIONS FOR APPROPRIATE COMPLEXITY ECOSYSTEM MODELLING

Authors:  Paul A. Moore, Maxwell C. Lukenbach, Dan K. Thompson, Nick Kettridge, Gustaf Granath, and James M. Waddington

Referee:  Anonymous Referee #1

Note:  Our response to referee comments are in red.

**General comments**

Moore et al. present a methodology to classify the peatland hummock-hollow variability for carbon flux modeling using high-resolution elevation models with k-means clustering. The study has collected samples across a variety of sites and mapped the high-resolution microtopography with the structure-from-motion [sic]. This manuscript provides insights into the influence of microtopography on the uncertainties of field sampling and carbon flux modeling. Considering the importance of the peatland carbon fluxes to the global climate change, this study is relevant and necessary. Overall, this manuscript is well-written and easy to follow. However, several issues in the manuscript need to be improved before the publication. This manuscript has done a nice analysis of the DEMs and evaluated its impacts on the NPP simulation. However, the validation of the generated DEM and the model-based fluxes is weak and needs to be strengthened. Another key issue for small scale flux simulation is to identify the optimal spatial resolution for modeling. I would also suggest the authors improve the analysis to identify the optimal spatial resolution to represent the microtopography. Generally, I think that this manuscript is publishable after revisions.

We kindly thank the referee for providing comments and constructive criticisms of the manuscript. We have added validation data for SfM-derived DEMs of hummock-hollow microtopography. These data have been added to the supplemental material as a short appendix and two new supplemental figures. In regards to the small scale flux modelling, the purpose of the empirical modelling was not to represent what the actual net photosynthesis of a given plot at a given site would be, but rather to highlight the potential bias introduced by modelling microtopography as a binary system With additional analysis, we have also added modelling results which examine how flux bias is affected by information loss (i.e. smoothing of DEM). Additional details are provided in responses to specific comments below.

**Specific comments**

1. L20: Some key words are repeating from the title. Normally key words should be different from words in the title, as to provide additional information.

Response: We have removed duplicate words and added a new key words (structure from motion, mire).

2. Abstract: The findings on the optimal spatial resolution is quite important for the appropriate complexity of flux modeling. As mentioned in L438-439, this manuscript concluded that on the optimal

resolution to represent the spatial variability has been identified from Fig. 2 and 5. These findings should be reflected in the Abstract.

Response: We have amended the abstract to include results on the appropriate scale of complexity to measure microtopographic variability at the site and plot scale. We have also produced additional modelling results of solar insolation net photosynthesis using progressively coarse smoothing functions applied to DEMs. Smoothed results are compared to end members (i.e. model output using unsmoothed DEMs and hummock-hollow binary approach) to provide a quantitative assessment of 'appropriate' model resolution.

3. Validation of the generated elevation model is needed. The structure-from-motion technique is sensitive to the camera geometric calibration, camera position information, and the accuracy and numbers of ground control points. Validation results on the generated DEM are necessary.

Response: We had previously done an analysis of the accuracy of the SfM technique as applied to our particular measurement design. We opted to omit these results, in part, to try and keep the manuscript more concise. We are happy to include our validation results but feel that they are most appropriate to include as part of the supplemental material (see Figures 1 and 2). We have also added a couple sentences to the 'Results: Digital elevation models of microtopography' section which includes basic summary statistics of SfM accuracy from our validation measurements. The validation results included in the revised supplemental material includes both laboratory and field validation measurements.  The primary purpose of laboratory measurements was to have greater accuracy and precision of x, y, z measurements. Our approach to validation was relatively simple and geared towards measurements of hummocks since the SfM approach itself has been well validated (e.g. Fonstad et al., 2013; Nouwakpo et al., 2014).

4. L237: Equations should be marked with a number. For the equation at L237, the variable x should be explained.

Response: We have added numbers to the equations and replaced 'x' in equation 2 with 'WC'.

5. L438-439: From Figure 5, it is not as easy as Figure 2 to identify the optimal resolution to represent the spatial variability. The authors can use additionally spatial analysis, e.g. semivariogram analysis, to strength these findings.

Response: We have opted to include the cumulative power spectral density in Figure 5 for consistency with Figure 2 and to provide objective information on the relative importance of scale for microtopographic variability at the plot scale.

6. The structure-from-motion can provide both DEMs and orthophotos for the study site. In the manuscript, the authors have used the DEMs for data analysis. Potentially, the orthophotos can be utilized to calculate vegetation indices to infer the vegetation growth conditions. This can improve the classification and NPP modeling. Why don't you make the best use of your data?

Response: In many cases, the actual moss species present in the plots do not match our choice of *Sphagnum* species for modelling net photosynthesis. While RGB information from SfM-derived orthophotos can certainly be used to help classify ground cover in peatlands (e.g. Harris and Baird, 2018), the purpose of the empirical modelling was not to represent what the actual net photosynthesis

of a given plot at a given site would be. Rather our purpose for using empirical models of net photosynthesis from the literature was to highlight the potential bias introduced by modelling microtopography as a binary system. Rather than focusing on inter-site species differences, the purpose of using multiple plots/sites in our analysis was primarily to include a variety of small-scale microtopographic distributions and not be biased to a particular site. Our choice of particular *Sphagnum* species to represent high-hummock, low-hummock, and hollow/lawn microtopographic classification is due to the observed niche partitioning along a microtopographic gradient presented in the literature (e.g. Andrus et al., 1983), and availability of empirical relations relating water content to water table depth, and net photosynthesis to water content in the published peer-reviewed literature.

7. Table 1: Mistakes on the Longitude. For instance, Sweden should not be -17W. I guess the fourth column should be the elevation above the mean sea level instead of the longitude.

Response: The site is located at 17 degrees longitude east, which is equivalent to -17 degrees longitude west. However, we have updated Table 1 longitudes to conform with SI standards so that positive longitude is degrees east, and negative longitude is degrees west.

8. Table 2: In Table 2, some plots have been classified into three members. However, we cannot see such three members in the histogram distribution of Figure 3 and 4. Please explain the reason.

Response: The GMM is representing the elevation distribution as a sum of Gaussian distributions. As such, a unimodal distribution that is skewed or is platy- or leptokurtic might be better represented by a multi-member Gaussian distribution than a normal distribution. While we feel that overlaying the GMM fits on Figures 3 and 4 would make them visually cluttered, we have opted to include a couple examples from our plots in the supplementary material. We have tried to include examples where the empirical distributions have clear separation in modes versus ones which don't. See Figure 3 below.

9. Figure S4. The curve should be from modeling and the dots are the measured one.

Response: Thanks for catching the error. We have updated Figure S4 accordingly.

10. Figure S8: the scale bar should be added into the spatial map. Otherwise, readers don't know the spatial scale of these maps.

Response: We have added a spatial scale to Figure S8.

11. The paper has done a nice analysis of the carbon flux modeling and assessed the impact of water table depth on the carbon fluxes. However, it is hard to evaluate whether these modeling is accurate enough or not. It would be better to add some chamber or eddy covariance measurements to validate the simulated NPP. Or at least compare the results with other relevant studies.

Response: Again, the purpose of the empirical modelling was not to represent what the actual net photosynthesis of a given plot at a given site would be, but rather to highlight the potential bias introduced by modelling microtopography as a binary system. However, we realised that it is not clear from the methods that the empirical models presented are from field-based studies of hummock-hollow plot-scale water content and flux measurements. We have revised the methods for clarity and also included the relevant source material, some of which was previously only cited in the figure captions. Moreover, we have added additional content to the discussion to compare the modelled net photosynthesis with other relevant studies.

**References:**

Andrus, R., Wagner, D., and Titus, J.: Vertical zonation of Sphagnum mosses along hummock-hollow gradients, Can. J. Bot., 61, 3128-3139, doi:10.1139/b83-352, 1983.

Fonstad, M. A., Dietrich, J. T., Courville, Jensen, J. L., and Carbonneau, P. E.: Topographic structure from motion: a new development in photogrammetric measurement, Earth Surface Processes and Landforms, 38(4), 421-430, doi: 10.1002/esp.3366, 2013.

Harris, A., and Baird, A. J.: Microtopographic Drivers of Vegetation Patterning in Blanket Peatlands Recovering from Erosion, Ecosystems, 1-20, doi: 10.1007/s10021-018-0321-6, 2018.

Nouwakpo, S. K., James, M. R., Weltz, M. A., Huang, C. H., Chagas, I., and Lima, L.: Evaluation of structure from motion for soil microtopography measurement, The Photogrammetric Record, 29(147), 297-316, doi: 10.1111/phor.12072, 2014.

**Figures:**

Figure 1: Spatial validation of structure-from-motion (SfM) method for lab (a-c) and field (d-f) microtopography. SfM reconstructions, manual measurements, and differences between the two are shown in the top, middle, and bottom panels, respectively.

[Figure]

Figure 2: Distribution of residuals between structure-from-motion (SfM) reconstruction and manual elevation measurements (a). Relation between magnitude of residuals and local slope (b). Results are bin averaged, where each point is based on 150, and 1000 measurements for the field and lab tests, respectively. Error bars indicate the standard error.

[Figure]

Figure 3: Gaussian mixture model (GMM) fit to relative frequency distribution of measured microtopographic elevation for four example plots. The full GMM distribution is obtained by summing the individual members. Examples for two- (upper panels) and three-member (lower panels) GMMs are given for elevation distributions which qualitatively show a separation of modes (left panels) versus ones where modes are not visually distinct (right panels).

[Figure]

---

## Author Comment (AC2) · 18 Jun 2019

**Reply to referee comments**

Manuscript:    ASSESSING THE PEATLAND HUMMOCK-HOLLOW CLASSIFICATION FRAMEWORK USING HIGH-RESOLUTION ELEVATION MODELS: IMPLICATIONS FOR APPROPRIATE COMPLEXITY ECOSYSTEM MODELLING

Authors:    Paul A. Moore, Maxwell C. Lukenbach, Dan K. Thompson, Nick Kettridge, Gustaf Granath, and James M. Waddington

Referee:    L. Kutzbach

Note:    Our response to referee comments are in red.

The manuscript of Moore et al. presents a very interesting and comprehensive analysis of the microtopographic structure of boreal non-patterned bogs. The paper scrutinizes the binary hummock-or-hollow classification approach, which is often followed in sampling design or modeling of biogeochemical and ecophysiological peatland processes.

The authors applied a well-designed combination of elaborate field data acquisition methods, targeted statistical analyses and appropriate process modeling. I am particularly pleased about the creative and thorough usage of various spatial statistical methods for analyzing the heterogeneity of peatland microtopography (e.g., Gaussian mixture models, Fourier transform power spectra of microtopographic variability along transects, slope and aspect analysis for microtopographic features, fractal dimension of plots). I also like the approach of simulating water content and net primary productivity in dependence of microtopography properties as an approach to demonstrate the relevance of thorough microtopography characterization for quantification of energy and matter fluxes. The authors show that non-consideration of the full continuum of microtopographical variability can lead to serious biases in spatial averages of net primary productivity due to negligence of microforms that are intermediate between hummocks and hollows. Even more pronounced bias would be expected for, e.g., methane emissions, which are controlled by water level depth below the moss surface in a highly nonlinear way.

Thus, the presented study is of high scientific relevance and originality. However, I think that the quality of the manuscript needs to be improved. In the following, I provide lists of (1.) general comments, (2.) specific comments, and (3.) technical comments. I recommend the manuscript for publication after major revisions.

General comments

(1) The experimental design of the study needs to be better explained. It is now too difficult for the reader to find out which method was applied where. That the many analyses were conducted at various peatland sites, needs to be more clearly stated already in the introduction. Furthermore, I think that a figure explaining the study design by including maps of different scale (e.g., northern hemisphere with location of all investigated peatlands, Nobel peatland with location of random plots in detail), would help. It would be also helpful if information on site and/or spatial scale would be added to all of the figure captions.

Response: In general, we used the terms 'site-level' and 'plot-level' to systematically orient the reader in methods/results. However, it is clear from the referee's comments that improved clarity is needed. As suggested, we have explicitly included 'site-level' or 'plot-level' to figure captions where appropriate for additional clarity. It is possible that this provides the necessary additional clarity, but we have also created a figure which provides visuals of the experimental design (see Figure 1 below). Given that the main manuscript already has nine figures and the size of the new figure, we feel that the new figure is best added to the supplemental material. However, we are happy to place it in the main text as is or in a modified form if there are any strong opinions on the matter

(2) The approach for modeling water content and potential NPP needs to be better described (L. 224-240). What is the basis for the parameterizations for water content for the different microforms? Please provide references. Is NPP considered as a CO2 flux or a carbon flux? Without specifying this, the modelled NPP values cannot be checked for plausibility. However, such a plausibility check would be necessary. Please compare your modelling results with empirical data on NPP of bog microforms.

Response: The purpose of the empirical modelling was not to represent what the actual net photosynthesis of a given plot at a given site would be, but rather to highlight the potential bias introduced by modelling microtopography as a binary system. However, we realised that it is not clear from the methods that the empirical models presented are from field-based studies of hummock-hollow plot-scale water content and capitula flux measurements. We have revised the methods for clarity and also included references to the relevant source material, some of which was previously only cited in the figure captions. Moreover, we have added additional content to the discussion to compare the modelled net photosynthesis with other relevant studies.

Specific comments

L. 50: I do not like this often used comparison because it is like comparing apples with oranges: The carbon pool of peatlands is estimated over their mean peat depth (can be more than 15 m), whereas carbon pools of soils are estimated for specific reference soil depths (e.g. 1 m , 3 m). Hence, do peatlands contain one third of the upper meter of global soils or of the upper 3 m or how many meters? Furthermore, soils store not only organic carbon but also inorganic carbon!

Response: Fair enough. We have removed the comparison from the introduction.

L. 69: I would think that the area covered by a hummock can be also quite larger than 1 m2.

Response: While we agree that hummocks can be quite larger than 1 $m^2$, we are trying to be somewhat general in the introduction and are referring to the order of magnitude (i.e. they are far more likely to be closer to 1 $m^2$ than 10 $m^2$). Nevertheless, we have softened the language to say that hummocks typically occupy and area of up to a few square meters.

L. 96: I suggest adding the reference: Cresto Aleina F., Runkle B. R. K., Kleinen T., Kutzbach L., Schneider J., Brovkin V. (2015): Modeling micro-topographic controls on boreal peatland hydrology and methane fluxes. Biogeosciences 12: 5689-5704.

Response: We appreciate the suggestion and have added the reference.

L. 112-113: Sentence not clear to me; please rewrite! I do not understand how you want to "explore DEM-derived properties" "using multi-site plot-scale sampling".

Response: We have revised the sentence which hopefully makes it clearer now.

L. 137: Write more specific: What kind of "individuals"? Have these been scientists, students, or farmers neighboring the peatland?

Response: We replaced "individuals" with "academic peatland researchers".

L. 157: Unit of resolution?

Response: We have updated to include the unit of resolution (i.e. pixels).

L. 234: According to SI system, do not mix units and quantities. Better "WC is the ratio of the mass of water and the mass of the non-water components of the soil (Unit: g g-1)."

Response: We have revised the sentence according to your suggestion.

L237: Specify the variable x. Probably, x equals WC, correct?

Response: Thanks for catching that. Yes, it is supposed to be WC and has been revised accordingly.

L 238: Better: ". . .represents percentage of maximum NPP"

Response: Revised accordingly.

L. 836: It is confusing to use the two terms "net photosynthesis" and "NPP" as y-axis titles of different diagrams in the same figure, respectively. Do you use the terms as synonyms? In my view, integration of net photosynthesis over time at the canopy scale leads to NPP; thus "net photosynthesis" and "NPP" would be closely related but not synonymous.

Response: We were admittedly a little sloppy with this abbreviation, where we used NPP to represent potential net photosynthesis. Understandably, this is easily confused with the widely used "net primary productivity", so we have replaced also cases of NPP in the manuscript by either spelling out "net photosynthesis" or abbreviating as NP.

Technical comments

Response: Where relevant for the technical comments, we have revised the manuscript according to the reviewer's comments/suggestions. Some suggestions were not adopted because the original text was removed as part of other revisions.

L. 29: Correct "examine" Done.

L. 31: Correct: "northern" Done.

L. 38: Correct: "positions" Done.

L. 50 Correct "one third"

Response: The text was removed as part of other revisions.

L. 107: Hyphenate: "plot-scale" Done.

L. 121: Hyphenate "transect-based" Done.

L. 145: I suggest writing: " 0.1 m x 0.1 m x 0.1 m (same for similar expressions throughout the manuscript) Done.

L. 179: Comma before "and" (beginning of independent sentence) Done.

L. 186: Number the equations. Done.

L. 208: better "selected" instead of "decided" Done.

L. 239: "mo" is not a standard abbreviation for a SI unit. Please define this somewhere before using it.

Response: We have opted to simply spell it out where used.

L. 296: I would move the F statistics in parentheses to the end of the sentence. Done.

L. 311: Infelicitous usage of statistical terminology: In my view, a result can be either significant or non-significant, give a specific error probability. It cannot be strongly of weakly significant.

Response: We agree that once a level of significance is chosen, that significance is determined by whether the p-value is equal to or less than the level of significance (i.e. reject null, results are significant) or greater than the level of significance (i.e. do no reject null, results are not significant). However, we also recognize that the choice of significance level is arbitrary to some degree, and that the p-value is an indicator of probability, so that the magnitude of the p-value could be interpreted as the null hypothesis being more/less probable on a continuous scale. The use of the terminology 'not strongly significant' was in part an attempt to recognize greater potential type II error given the sample size and p-value near the significance level. Nevertheless, we have opted to switch the statement to 'not significant'.

L. 374: Hyphenate: "under-samples" Done.

L. 380: Better a full stop instead of a comma after "conditions"

Response: Unfortunately, because "conditions" was used twice on line 380 of the submitted manuscript, I'm not sure which "conditions" you were referring to.

L. 465: Comma before "which" Done.

L. 507: Hyphenate "water table-dependent" Done.

L. 516: Comma before "where" Done.

L. 532: Comma before "where" Done.

Figure 1: Overview of site locations, site-level measurement design, and plot-level hummock-hollow pairs (see Table 1 for additional details).

Site locations

[Figure]

Red Earth Creek – site-level analysis

[Figure]

DEMs of select plot-level hummock-hollow pairs